# Genetic Overlap between General Cognitive Function and Schizophrenia: A Review of Cognitive GWASs

**DOI:** 10.3390/ijms19123822

**Published:** 2018-11-30

**Authors:** Kazutaka Ohi, Chika Sumiyoshi, Haruo Fujino, Yuka Yasuda, Hidenaga Yamamori, Michiko Fujimoto, Tomoko Shiino, Tomiki Sumiyoshi, Ryota Hashimoto

**Affiliations:** 1Department of Neuropsychiatry, Kanazawa Medical University, Uchinada, Ishikawa 920-0293, Japan; ohi@kanazawa-med.ac.jp; 2Medical Research Institute, Kanazawa Medical University, Ishikawa 920-0293, Japan; 3Faculty of Human Development and Culture, Fukushima University, Fukushima 960-1296, Japan; sumiyoshi@educ.fukushima-u.ac.jp; 4Graduate School of Education, Oita University, Oita 870-1192, Japan; fjinoh@hus.osaka-u.ac.jp; 5Department of Pathology of Mental Diseases, National Institute of Mental Health, National Center of Neurology and Psychiatry, Tokyo 187-8553, Japan; yasuda@psy.med.osaka-u.ac.jp (Y.Y.); yamamori@psy.med.osaka-u.ac.jp (H.Y.); tshiino@ncnp.go.jp (T.S.); 6Department of Psychiatry, Osaka University Graduate School of Medicine, Suita, Osaka 565-0871, Japan; mfujimoto@psy.med.osaka-u.ac.jp; 7Department of Preventive Interventions for Psychiatric Disorders, National Institute of Mental Health, National Center of Neurology and Psychiatry, Kodaira, Tokyo 187-8553, Japan; sumiyot@ncnp.go.jp; 8Osaka University, Suita, Osaka 565-0871, Japan

**Keywords:** schizophrenia, general cognitive function, intelligence, GWAS, genetic correlation

## Abstract

General cognitive (intelligence) function is substantially heritable, and is a major determinant of economic and health-related life outcomes. Cognitive impairments and intelligence decline are core features of schizophrenia which are evident before the onset of the illness. Genetic overlaps between cognitive impairments and the vulnerability for the illness have been suggested. Here, we review the literature on recent large-scale genome-wide association studies (GWASs) of general cognitive function and correlations between cognitive function and genetic susceptibility to schizophrenia. In the last decade, large-scale GWASs (*n* > 30,000) of general cognitive function and schizophrenia have demonstrated that substantial proportions of the heritability of the cognitive function and schizophrenia are explained by a polygenic component consisting of many common genetic variants with small effects. To date, GWASs have identified more than 100 loci linked to general cognitive function and 108 loci linked to schizophrenia. These genetic variants are mostly intronic or intergenic. Genes identified around these genetic variants are densely expressed in brain tissues. Schizophrenia-related genetic risks are consistently correlated with lower general cognitive function (*r_g_* = −0.20) and higher educational attainment (*r_g_* = 0.08). Cognitive functions are associated with many of the socioeconomic and health-related outcomes. Current treatment strategies largely fail to improve cognitive impairments of schizophrenia. Therefore, further study is needed to understand the molecular mechanisms underlying both cognition and schizophrenia.

## 1. Introduction

Cognitive functions play important roles in mental and physical well-beings. This is supported by observations that people with higher intelligence tend to have greater educational attainment, more professional jobs, higher incomes, and increased longevity [1,2]. Accordingly, impairments of cognitive functions result in social and occupational dysfunction which leads to poor life outcomes [3,4,5,6,7,8].

Cognitive disturbances are a core feature of schizophrenia—a psychiatric disorder with clinical and genetic heterogeneity [9,10]. Compared with healthy individuals, patients with schizophrenia demonstrate about a 1–2 standard deviation decline in performance on tests of several cognitive domains, including working, verbal and visual memories, processing speed, attention, social cognition, and intelligence [11,12,13,14,15,16,17]. Although the disorder is generally characterized by positive (e.g., hallucinations and delusions) and negative (blunted affect and withdrawal) symptoms, cognitive impairments should also be considered as an independent clinical dimension [18,19]. These impairments exist before the onset of illness and are worsened around it [20,21,22]. It has been suggested that cognitive deficits of schizophrenia may be resistant to treatment with antipsychotic drugs [23,24,25,26]. This indicates a need for clarifications of the mechanisms underlying these conditions.

Schizophrenia has a strong genetic basis with an estimated heritability of approximately 80% [27]. Cognitive functions such as general intelligence also have a genetic component (*h*^2^ = 0.33–0.85) [28,29,30,31,32]. Despite the difference in heritability for intelligence between childhood (*h*^2^ = 0.45) and adulthood (*h*^2^ = 0.80), there is a high correlation between IQ levels in childhood and those in adulthood (*r_g_* = 0.89) [33]. Relatives or twin siblings of patients with schizophrenia have also displayed impaired cognitive function to a lesser extent [9,34]. These findings suggest the contribution of genetic components to cognitive impairments in schizophrenia.

Genome-wide association studies (GWASs) that examine millions of genetic variants are a powerful tool to identify common variants responsible for susceptibility to common and complex diseases. The largest GWAS to date is the Psychiatric Genomics Consortium (PGC) using 36,989 patients with schizophrenia and 113,075 controls, which has identified 108 loci including genes and genetic variants related to schizophrenia [35]. Several consortia, such as the Cognitive Genomics Consortium (COGENT), Heart and Aging Research in Genomic Epidemiology Consortium (CHARGE), and UK Biobank (UKB), have performed GWASs to identify genetic loci related to cognitive function [36,37,38,39,40,41,42,43]. GWASs with fewer than 20,000 subjects did not find any significant loci [36,44,45,46]. These GWAS consortia used diverse assessment tools to represent targeted cognitive constructs in various samples, e.g., general cognitive function (*g*), Intelligence Quotient (IQ), fluid intelligence, etc., which could have been subject to phenotypic heterogeneity. By contrast, GWASs using samples from nearly 300,000 individuals successfully detected more than 100 genome-wide significant loci related to cognitive function [42,43]. In addition, part of the genetic correlation in the genetic effects identical between cognitive function and schizophrenia has been identified [38,39,40,41,42]. Therefore, cognitive functions have been proposed as an intermediate phenotype or biotype [9,15,47,48] to explain the mechanisms involved in the pathogenesis of schizophrenia.

In this article, we review the literature on recent large-scale GWASs of general cognitive function and genetic correlations between cognitive function and schizophrenia.

## 2. General Cognitive Function (*g*)

A number of tests have been used to measure various domains of cognitive functions. It is difficult to perform GWASs of cognitive functions uniformly because these cognitive tests vary among study cohorts. Twin and family studies show strong genetic correlations across diverse cognitive domains [49]. Under this circumstance, general cognitive function (*g*) is defined as a latent trait underlying shared variance across multiple subdomains of cognition [36,37,39,44,45]. To extract *g*, principal component analysis (PCA) is required on at least one cognitive measure across at least three domains, e.g., logical memory for verbal declarative memory, digit span for working memory, and digit symbol coding for processing speed. In other words, the first unrotated principal component of several distinct neuropsychological tests is obtained from the PCA. For example, an average of eight neuropsychological tests across COGENT cohorts were selected: digit span, digit symbol coding, verbal memory for words, visual memory, semantic fluency, word reading, verbal memory for stories, phonemic fluency, vocabulary, and the trail-making test [39]. The first principal component obtained accounted for approximately 40% of the variance in overall test performance. The *g* factors extracted from different cognitive tests were strongly correlated (>0.98) [50], supporting the universality of *g*.

Several cognitive GWASs have been performed using the *g* approach [15,36,37,39,44,45]. GWASs with fewer than 20,000 subjects did not find any genome-wide significant variants [15,36,44,45], while the GWASs with 35,298 [39] and 53,949 [37] subjects successfully identified two (*RP4-665J23.1* on 1p22.2 and *CENPO* on 2p23.3) and three (*MIR2113* on 6q16.1, *AKAP6/NPAS3* on 14q12 and *TOMM40/APOE* on 19q13.32) genome-wide significant loci, respectively (Table 1). However, neither these loci, nor the reproducibility of the findings, were consistent across studies.

## 3. Fluid Intelligence

Fluid-type intelligence requires swift thinking, relies relatively little on prior knowledge, and is often measured by unfamiliar and sometimes abstract materials [44]. By contrast, crystallized-type intelligence is typically assessed using tests such as those for acquired knowledge and vocabulary [44]. The discrepancy between fluid and crystallized intelligence becomes particularly noticeable in late adulthood—the age-related decline of fluid intelligence comes earlier and more rapidly [51,52].

To assess crystallized intelligence, either the National Adult Reading Test or the WAIS vocabulary subtest is used. Fluid intelligence, which may be equivalent to *g*, is assessed using PCA of data from several cognitive tests, such as logical memory, verbal fluency, auditory verbal learning tests (AVLT), and subtests from the Wechsler Adult Intelligence Scale (WAIS)-III [44]. Fluid intelligence is also measured by the verbal–numerical reasoning (VNR) test [42]. This test uses 13 multiple-choice questions—six verbal and seven numerical—which are presented on a touchscreen computer in either an assessment center or a web-based format at home [38,40]. Scores are obtained from the number of questions answered correctly in two minutes. With this method, the GWAS in UKB (*n* = 36,035) detected three genome-wide significant loci, including several genes, e.g., *CYP2D6* and *NAGA* at 22q13.2, *FUT8* at 14q23.3 and *PDE1C* at 7p14.3 [38].

Because performance on the VNR is correlated with *g* [40,53], the level of power in recent GWASs has been increased through combinations of *g* and fluid intelligence [40,41,42,43]. The total sample sizes in these studies were approximately 80,000–300,000 (Table 1). For example, one of the recent GWASs with 269,867 subjects identified 205 genome-wide significant loci [43]. This GWAS also identified some overlapping loci (2p23.3, 6q16.1, 7p14.3, 14q12, 19q13.32, and 22q13.2) consistent with previous reports [37,38,39], although these loci did not fully include lead genetic variants. The sample size in GWASs is positively correlated with the number of genome-wide associated loci detected (Figure 1, *r*^2^ = 0.92, *p* = 1.18 × 10^−5^). Several of these loci overlapped with those associated with schizophrenia, such as 1p21, 1p34, 2q24, 2q33, 3p21, 3q22, 4q24, 5q21, 6p22, 7q22, 8q24, 11q25, 12q24, 14q12, 14q32, 16q22, and 22q13 [35,41,43].

## 4. Educational Attainment

Educational attainment, represented by the number of years of education, is strongly influenced by genetic and environmental factors [54,55]. At least 20% of the variation among individuals is accounted for by genetic factors [54]. GWASs of educational attainment in 111,114 and 293,723 European individuals identified 14 genome-wide significant loci associated with the attainment of a college or university degree [38] and 74 loci associated with the number of years of schooling completed [55], respectively. Individuals with a higher level of intelligence tend to stay in school longer and attain higher qualifications than those with a lower level of intelligence. In addition, general cognitive ability (fluid intelligence) is correlated with educational attainment (*r_g_* > 0.70) [38,39,40,41,43]. Therefore, educational attainment is useful as a proxy phenotype for general cognitive function in GWAS analyses. In fact, several loci, such as 1p31.1, 2q11.2, 3p21.31, 6q16.1, and 13q21.1, in a GWAS of educational attainment overlapped with those of general cognitive function.

## 5. Genes and Functions Related to General Cognitive Function

The genetic variants related to general cognitive function were mostly intronic or intergenic. The genes identified around these genetic variants were densely expressed in the brain [42,56], specifically striatal medium spiny neurons and hippocampal pyramidal CA1 neurons [43]. Common gene functions linked to general cognitive function were determined in gene-set analyses in some GWASs [40,42,43]. These functions include neurogenesis, regulation of nervous system development, neuronal differentiation, and regulation of cell development. Functions such as neuron projection and regulation of synaptic structure/activity were also associated with general cognitive function. As pathways related to these functions have been implicated in the pathophysiology for general cognitive function, these findings suggest that brain-expressed genes contribute to general cognitive function via neurodevelopmental processes in specific brain cells.

Smeland et al. (2017) extensively investigated shared genetic loci of the GWAS by conditional false discovery rate analysis and identified 21 genomic loci jointly influencing cognitive functions and vulnerability to schizophrenia [56]. Of the 21 loci, 18 showed a negative correlation between risk of schizophrenia and cognitive performance. The locus most strongly shared was detected on 22q13.2 that contains *TCF20*, *CYP2D6*, and *NAGA*. In addition, this locus was shown to have quantitative trait locus (eQTL). *NAGA* encodes lysosomal enzymes that modify glycoconjugates, and *CYP2D6* encodes cytochrome P450 enzymes that metabolize a broad range of drugs [56]. Other loci, including *KCNJ3*, *GNL3* and *STRC*, were also identified as eQTLs. Although these genes shared by two phenotypes are not localized in specific pathways, they may provide potential drug targets for improving cognitive impairments in patients with schizophrenia.

## 6. Polygenic Risk Score Analysis and Genetic Correlation between General Cognitive Function and Schizophrenia

Polygenic overlaps between alleles of general cognitive function and schizophrenia risk have been examined [36,57]. On the basis of the polygenic risk scores (PRS) derived from GWASs, a set of alleles associated with lower general cognitive function predicted an increased risk of vulnerability to schizophrenia. Conversely, polygenic alleles associated with schizophrenia-related risks predicted lower cognitive functions—particularly general cognitive function, performance IQ, attention, and working memory [36,57,58,59,60,61,62,63]. Thus, greater PRS related to risks for schizophrenia were associated with a greater decline in IQ after childhood in the general population [58]. So far, most studies on cognitive functions have used general population [36,57,58,60,61,62,63], and have not been specific to patients with schizophrenia [59,64].

Linkage disequilibrium score regression (LDSC) analysis estimates genetic variant correlations (*r_g_*) from GWASs and is a powerful tool for investigating genetic architectures of common traits and diseases [65]. Studies using this method have consistently reported negative correlation between general cognitive function and schizophrenia-related risks, with *r_g_* of approximately −0.2 (Figure 2) [39,40,41,42,43]. Specifically, higher educational attainment is associated with lower schizophrenia risk [66], whereas lower educational attainment predicts worse premorbid function and poorer outcomes [66]. These correlations would be reasonable in view of positive correlations between educational attainment and general cognitive function (Figure 3). However, recent studies found a positive correlation between educational attainment and schizophrenia (Figure 2) [55,67]. This discrepancy may be explained by at least two disease subtypes, i.e., patients with high intelligence, and those with cognitive impairments [68].

## 7. Genetic Correlations between General Cognitive Function and Socioeconomic and Health-Related Outcomes

Cognitive function has been shown to be correlated with socioeconomic and health-related outcomes as well as neuropsychiatric disorders, as evidenced by LDSC analysis (Figure 3) [39,40,41,42,43]. Educational attainments provide the most robust correlations among other phenotypes. Specifically, better cognitive function was associated with a lower risk of several neuropsychiatric disorders, including schizophrenia, major depressive disorder, bipolar disorder, attention-deficit hyperactivity disorder, anxiety disorder, and Alzheimer’s disease. By contrast, a higher risk of autism spectrum disorder was related to better cognitive function. As individuals around adolescence were included in correlational analyses (Table 1), the timing of cognitive assessment, i.e., before or after onset of the illness, may have obscured the results from these analyses.

Better cognitive function was associated with lower levels of neuroticism, depressive symptoms, and insomnia (Figure 3). Physical factors contributing were smaller waist-to-hip ratio and waist circumference, smaller volume of putamen, fewer numbers of cigarettes per day, less likelihood of having ever smoked, and lower body mass index in adulthood. Other factors affecting cognition included fewer children, higher levels of openness, age of onset of smoking and smoking cessation, larger intracranial volume, larger head circumference in infancy, height, birth length and weight, higher age of first birth, and greater longevity. These findings indicate that general cognitive function is related to socioeconomic and health-related outcomes in addition to neuropsychiatric disorders.

## 8. Intelligence Decline in Schizophrenia

Intelligence decline is conceptualized as intra-individual difference in intellectual performance between different time points [18,47,48,69]. Thus, it may be calculated by subtracting estimated premorbid IQ, as measured by the Adult Reading Test, and the present IQ, as measured by the WAIS. For the purpose of brief assessment, we have recently developed the WAIS-Short Form consisting of the Similarities and Symbol Search subtests [70]. Because clinical trials targeting cognitive impairment of schizophrenia have mostly yielded negative results, we suggest that patients without intelligence decline be excluded from participation. To date, no large-scale GWAS for intelligence decline in patients with schizophrenia has been performed, and further studies are needed.

The degree of intelligence decline in patients with schizophrenia is typically classified into three intellectual levels [18,23,47,69,71,72,73,74,75,76,77]:(a)Deteriorated group: patients with a difference of 10 points or more between premorbid IQ and present IQ;(b)Preserved group: patients with a difference of less than 10 points between premorbid IQ and present IQ (premorbid IQ above 90);(c)Compromised group: patients with a difference of less than 10 points between premorbid IQ and present IQ (premorbid IQ below 90).

The compromised IQ subgroup includes patients who have intellectual disability. Although cognitive impairments are a core feature of schizophrenia, approximately 30% of patients are classified into the preserved IQ subgroup [47].

So far, GWAS, PRS, or LDSC analysis has not been performed based on the above classification (deteriorated, preserved, and compromised IQ) in patients with schizophrenia. As the current diagnostic criteria for schizophrenia is independent of cognitive traits and genetic architectures, GWASs based on intelligence decline subgroups may reveal novel genetic variants specific to cognitive impairments. Caution is needed in interpreting data from IQ measures, as they are subject to non-specific consequences of schizophrenia, effects of medication, and cognitive decline preceding the onset of illness. Additionally, IQ scores by themselves cannot describe specific cognitive domains that are relatively more affected than others in individual patients.

## 9. Conclusions

In this paper, we reviewed the literature of recent large-scale GWASs targeting general cognitive function, a phenotype that captures shared variations in performance on tests of several cognitive domains. Studies on polygenic correlations between cognitive function and schizophrenia were also addressed. In the last decade, large-scale GWASs have identified more than 100 loci linked to general cognitive function and schizophrenia. Genetic variants identified are mostly intronic or intergenic, and genes around them are densely expressed in brain tissues. Substantial proportions of the heritability of these phenotypes are explained by polygenic architectures consisting of many genetic variants with small effects. General cognitive function has been reported to be genetically correlated with socioeconomic and health-related outcomes, as well as neuropsychiatric disorders. In particular, lower general cognitive function has been consistently correlated with schizophrenia risks. Current treatment strategies largely fail to improve cognitive impairments of schizophrenia. In order to progress, further study is needed to understand the shared pathogenesis for general cognitive function in relation to the illness.

## Figures and Tables

**Figure 1 ijms-19-03822-f001:**
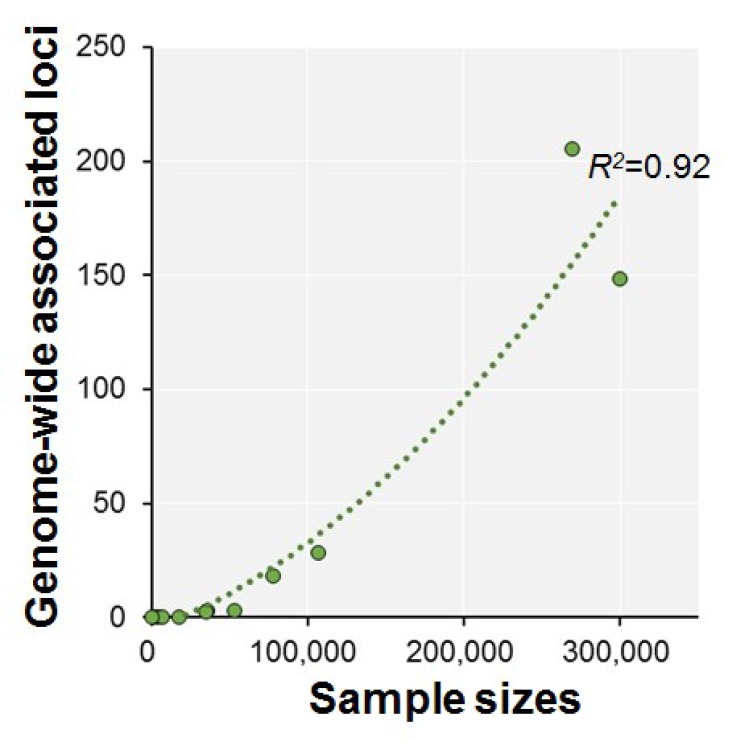
Relationship between sample sizes in GWASs of general cognitive function and genome-wide associated loci detected in each GWAS. Circles represent GWASs.

**Figure 2 ijms-19-03822-f002:**
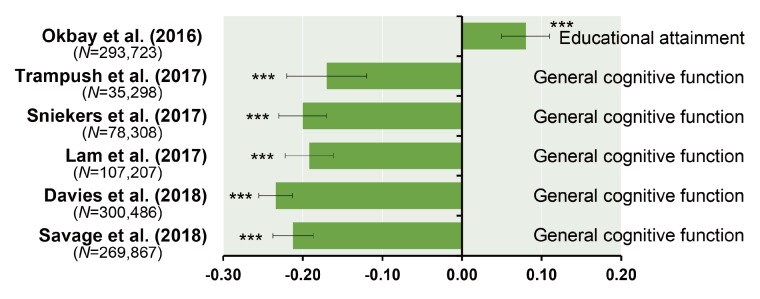
Genetic correlations (*r_g_*) of educational attainment or general cognitive function with schizophrenia. Error bars indicate the SE of *r_g_*. *** *p* < 0.001.

**Figure 3 ijms-19-03822-f003:**
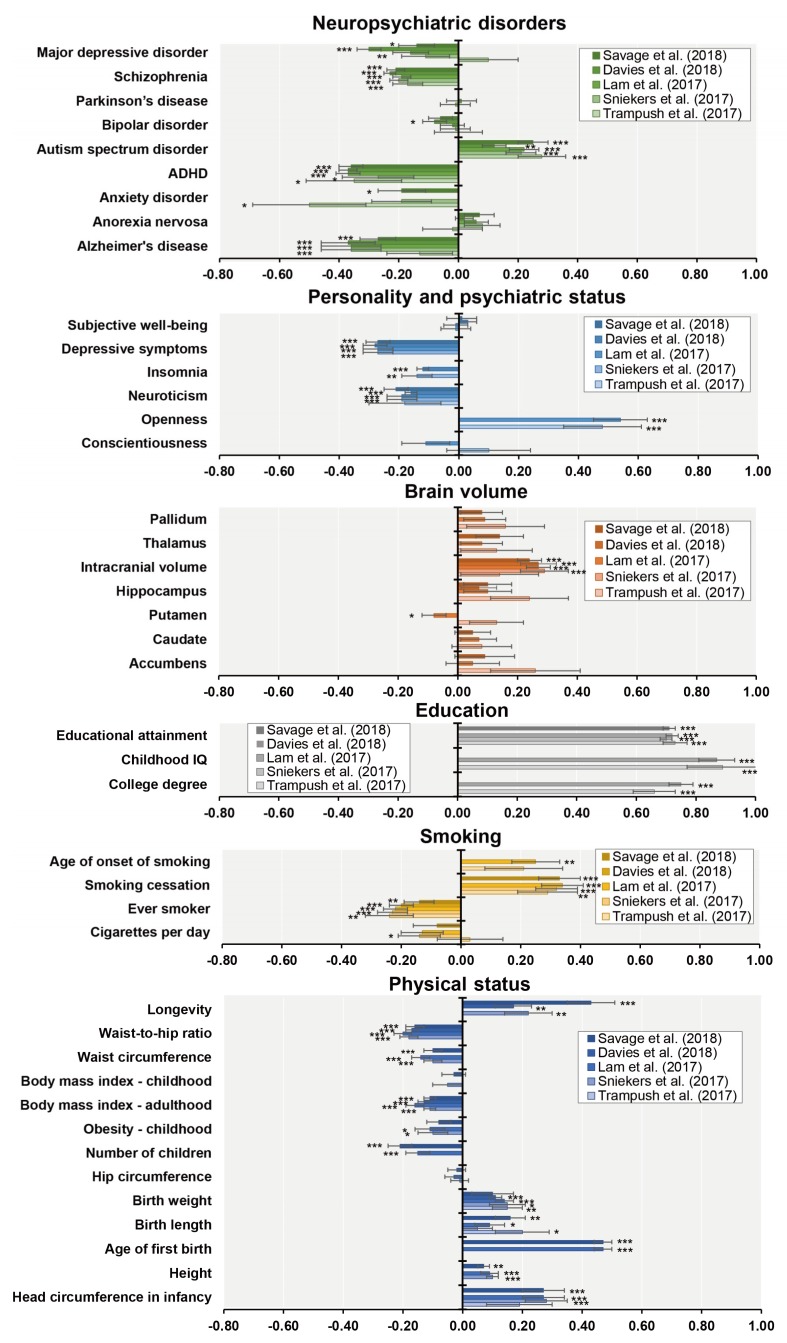
Genetic correlations (*r_g_*) between general cognitive function and several phenotypes. Error bars indicate the SE of *r_g_*. * *p* < 0.05, ** *p* < 0.01, *** *p* < 0.001.

**Table 1 ijms-19-03822-t001:** A summary of GWASs of general cognitive function.

Authors (year)	*n*	Phenotypes	Ethnics	Participants	Consortium	Age Range	GWS Loci	SNP Hits	GWS Gene
Ohi et al. (2015) [15]	411	*g* or IQ	Japanese	Psychiatric healthy subjects	Osaka University	18–66	0	0	NA
Davies et al. (2011) [44]	3511	*g*	Caucasian	Nonclinical healthy samples	CAGES, LBC1921, LBC1936, ABC1936, etc.	64.6–79.1 *	0	0	1
Lencz et al. (2014) [36]	5000	*g*	Caucasian	General population (epidemiologically representative cohorts or mentally healthy cohorts)	COGENT	15.9–69.5 *	0	0	NA
Benyamin et al. (2014) [45]	17,989	*g* or IQ	European	Children	CHIC	6–18	0	0	0
Kirkpatrick et al. (2014) [46]	7100	IQ	Caucasian	Community-based family study samples	MTFS, SIBS	11.8–43.3 *	0	0	0
Davies et al. (2015) [37]	53,949	*g*	European	Population-based cohorts	CHARGE	>45	3	13	1
Davies et al. (2016) [38]	36,035	Fluid intelligence (VNR)	White British	Touchscreen-based community-dwelling individuals	UKB	40–73	3	149	17
Trampush et al. (2017) [39]	35,298	*g*	European	General population	COGENT	8–96	2	7	7
Sniekers et al. (2017) [40]	78,308	*g*, IQ or Fluid (VNR)	European	Web-base and touchscreen-based community-dwelling individuals and population-based cohorts	UKB, CHIC, MTFS, etc.	8–78	18	336	47
Lam et al. (2017) [41]	107,207	*g*, IQ or Fluid (VNR)	European	Web-base and touchscreen-based community-dwelling individuals and population-based cohorts	COGENT, UKB, CHIC, etc.	8–96	28	469	73
Davies et al. (2018) [42]	300,486	*g* or Fluid (VNR)	European	Web-base and touchscreen-based community-dwelling individuals and population-based cohorts	CHARGE, COGENT, UKB	16–102	148	11,600	709
Savage et al. (2018) [43]	269,867	*g*, IQ or Fluid (VNR)	European	Epidemiological cohorts	COGENT, UKB, etc.	5–98	205	12,110	507

VNR, Verbal–numerical reasoning; CAGES, Cognitive Aging Genetics in England and Scotland; LBC1921, LBC1936, Lothian Birth Cohorts of 1921 and 1936; ABC1936, Aberdeen Birth Cohort 1936; COGENT, Cognitive Genomics Consortium; MTFS, Minnesota Twin Family Study; SIBS, Sibling Interaction & Behavior Study; CHARGE, Heart and Aging Research in Genomic Epidemiology consortium; UKB, UK Biobank; GWS, Genome-wide significant. * Mean age range among cohorts was indicated.

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
