# Peer review of "Genetic Overlap between General Cognitive Function and Schizophrenia: A Review of Cognitive GWASs"

_ijms, 2018, doi:10.3390/ijms19123822_

Reviewer 1 Report

This review by Ohi and colleagues first aims to summarize the GWAS literature related to cognitive functions and schizophrenia and second to review the existing evidence for a genetic overlap between these two phenotypes. The review is well written and the authors have used the extant literature. My comments are below:

What I would have liked to see in this review is a better integration followed by a more intuitive coordination between their description of the GWAS for cognitive functions and schizophrenia. While they state that there is a genetic overlap between the two phenotypes, they have not provided a more tangible link from the existing literature to support that. For example, it would have been useful if the authors have provided more information and/or discussion about the potential biological functions of the loci which were used to generate the PRS. This can provide a more biological narrative to explain the genetic link between a reduced cognitive functions and schizophrenia.   

I also believe, that the review would have benefitted from a more thorough discussion regarding the potential treatment of cognitive decline in patients with schizophrenia. For example, how the known function of the genes that might be implicated in both phenotypes may inform us to develop a better treatment or diagnostic plans.

The review would have benefitted from a more clear direction. At the moment, at least to me, it reads more like an abstract compilations from the cited literature.   

Author Response

Response to Reviewer 1

Comment 1:

The reviewer suggested providing more information and/or discussion about the potential biological functions of the overlapped loci between cognitive functions and schizophrenia. According to the reviewer’s suggestion, we have mentioned it as follows:

P6, L28-39

Smeland et al. (2017) has extensively investigated shared genetic loci of the GWAS for cognitive functions and schizophrenia by conditional false discovery rate analysis [58]. The study identified 21 genomic loci jointly influencing cognitive functions and schizophrenia. Eighteen of 21 loci showed a negative correlation between risk of schizophrenia and cognitive performance. The strongest shared locus was detected on 22q13.2 that contains many genes, such as TCF20, CYP2D6 and NAGA. In addition, expression quantitative trait locus (eQTL) was detected for the same 22q13.2 locus in brain tissues; the identified genetic variants related to cognitive functions and schizophrenia regulate CYP2D6 and NAGA expression in the human brain. NAGA encodes a lysosomal enzyme that modifies glycoconjugates, and CYP2D6 encodes a cytochrome P450 enzyme that metabolizes a broad range of drugs [58]. Other loci including several genes, KCNJ3, GNL3 and STRC, were also detected as eQTL. Although genes in the loci shared between two phenotypes were not enriched for specific pathways, these genes may be potential drug targets for improving cognitive impairments in patients with schizophrenia.

Comment 2:

The reviewer suggested discussing the potential treatment of cognitive decline in patients with schizophrenia. According to the reviewer’s suggestion, we have discussed it as follows:

P6, L28-39

Smeland et al. (2017) has extensively investigated shared genetic loci of the GWAS for cognitive functions and schizophrenia by conditional false discovery rate analysis [58]. The study identified 21 genomic loci jointly influencing cognitive functions and schizophrenia. Eighteen of 21 loci showed a negative correlation between risk of schizophrenia and cognitive performance. The strongest shared locus was detected on 22q13.2 that contains many genes, such as TCF20, CYP2D6 and NAGA. In addition, expression quantitative trait locus (eQTL) was detected for the same 22q13.2 locus in brain tissues; the identified genetic variants related to cognitive functions and schizophrenia regulate CYP2D6 and NAGA expression in the human brain. NAGA encodes a lysosomal enzyme that modifies glycoconjugates, and CYP2D6 encodes a cytochrome P450 enzyme that metabolizes a broad range of drugs [58]. Other loci including several genes, KCNJ3, GNL3 and STRC, were also detected as eQTL. Although genes in the loci shared between two phenotypes were not enriched for specific pathways, these genes may be potential drug targets for improving cognitive impairments in patients with schizophrenia.

P8, L96-110

Because current treatment strategies including antipsychotics and cognitive remediation as well as clinical drug trials largely fail to improve the cognitive impairments in patients with schizophrenia, we suggest that patients without intelligence decline should be initially detected and excluded from clinical drug trials meant to improve cognitive impairments in patients with schizophrenia.

Comment 3:

The reviewer suggested indicating a clearer direction of this review. According to the reviewer’s suggestion, we have indicated it in abstract as follows:

P1, L38-42

Because current treatment strategies largely fail to improve the cognitive impairments in patients with schizophrenia, further study is needed to understand the shared molecular biological mechanisms underlying general cognitive function and schizophrenia and to develop more effective treatment for cognitive impairments in schizophrenia.

Reviewer 2 Report

This manuscript by R. Hashimoto and colleagues is a timely summary of the genetic correlates of cognition and schizophrenia. As cognitive declines are part of schizophrenia and also a predictor for schizophrenia onset among risk carriers, the topic is highly significant. The manuscript would be improved by incorporating the following specific points:

1) The term "proxy genes" is confusing.  

2) Please quantitatively define "preferably" when describing expression of "proxy" genes in the brain.

3) Page 2, line 54-55: This sentence should be re-written.

4) Page 2, line 56-67: This statement is inconsistent with a later statement (Page 8, section 8).

5) Page 3, lines 107-111: If identified genes differ among studies, reproducibility is not good.

6) Page 6,line 30:" in the reverse" is not clear.

7) Page 6, line 46: This sentence should be re-written "individuals with ...tend to be more education".

8) Page 6, lines 44-50: this statement is not clear. My take of these data is that educational attainment has very weak correlation with schizophrenia risk(i.e., less than 0.1) but general cognitive functions are negatively correlated with schizophrenia.

9) It is not clear why Fig 2 is needed. The data of Okbay et al (2016) can be included in Fig 3, Education, Educational Attainment.

10) Fig 3. Please provide some statistical values to indicate whether correlations are meaningfully significant or just non-significant trends. Moreover, the authors should describe when cognitive functions were measured in relation to the onset of psychiatric disorders.

11) The term "ID" is the standard abbreviation of intellectual disability. The authors are advised to avoid their idiosyncratic use of ID as an abbreviation of intellectual decline. This reviewer does not find it necessary to abbreviate "intellectual decline".

12) A GWAS analysis of n=166 is practically meaningless. Such data should not be discussed.

13) IQ measures after the onset of schizophrenia include a non-specific consequence of schizophrenia or effects of medication and a mechanistically-linked cognitive decline preceding the onset of illness. This point should be made clearer.

14) IQ scores might not be sensitive enough to detect deficits in specific cognitive domains that are more affected in schizophrenia than other domains. The observation that "preserved IQ" group has schizophrenia without cognitive declines should be interpreted with caution.

Author Response

Response to Reviewer 2

Comment 1:

The reviewer pointed out that the term ‘proxy genes’ was confusing. According to the reviewer’s suggestion, we have avoided using the term as follows:

P1, L34-35

The identified genes around these genetic variants are strongly expressed in brain tissues.

Comment 2:

The reviewer suggested defining quantitatively ‘preferably’ when describing expression of ‘proxy’ genes in the brain. According to the reviewer’s suggestion, we have used term ‘strongly’ but not ‘preferably’ thorough manuscript.

Comment 3:

The reviewer suggested rewriting the sentence ‘Cognitive impairments are exhibited around or after disorder onset, but impairments exist even before onset [20-22]’. According to the reviewer’s suggestion, we have revised it as follows:

P2, L57-58

Cognitive impairments exist before disorder onset, and the impairments are emphasized around or after onset [20-22].

Comment 4:

The reviewer pointed out that the sentence ‘Cognitive impairments are stable over time though partly affected by antipsychotic medications [23-26]’ was inconsistent with a later statement (Page 8, section 8). As the referee’s pointed out, these statements were inconsistent. We have revised the sentence as follows:

P2, L59-60

Cognitive impairments are stable over time after onset, and poorly affected by current antipsychotic medications [23-26].

Comment 5:

Page 3, lines 107-111: ‘Using the g approach, several cognitive GWASs have been performed [15,38,39,41,46,47]. GWASs with fewer than 20,000 subjects did not find any genome-wide significant variants [15,38,46,47], while GWASs with 35,298 [41] and 53,949 [39] subjects successfully identified two (RP4-665J23.1 on 1p22.2 and CENPO on 2p23.3) and three genome-wide significant loci (MIR2113 on 6q16.1, AKAP6/NPAS3 on 14q12 and TOMM40/APOE on 19q13.32), respectively (Table 1). These identified loci differed between the studies.’ The reviewer pointed out that if identified genes differ among studies, reproducibility is not good. We agree with the reviewer’s opinion. We have revised the sentence as follows:

P3, L113-114

However, these identified loci differed between the studies, and reproducibility was not good.

Comment 6:

The reviewer suggested clarifying ‘the reverse’ in the sentence ‘Polygenic overlaps between alleles of general cognitive function and schizophrenia risk were examined, in addition to the reverse [38,59].’. According to the reviewer’s suggestion, we have clarified it as follows:

P6, L42-44

Polygenic overlaps between alleles of general cognitive function and schizophrenia risk and the reverse polygenic overlaps between alleles of schizophrenia risk and general cognitive function were examined [38,59].

Comment 7:

The reviewer suggested rewriting the sentence ‘individuals with higher cognitive function in childhood and adolescence tend to be more education’. According to the reviewer’s suggestion, we have revised it as follows:

P6, L59-60

individuals with higher cognitive function in childhood and adolescence tend to have longer years of education

Comment 8:

‘Considering the strong positive genetic correlation between general cognitive function and educational attainment (Figure 3) and the evidence that individuals with higher cognitive function in childhood and adolescence tend to be more education, progress to more professional and better-paid jobs, live healthier lives, and live longer, these findings suggest that genetic effects that contribute to educational attainment but not via cognitive function may be responsible for the observed positive genetic correlation between educational attainment and schizophrenia [70].’ The reviewer suggested clarifying the sentences. According to the reviewer’s suggestion, we have clarified it as follows:

P6, L58-64

Despite the strong positive genetic correlation between general cognitive function and educational attainment (Figure 3) and the evidence that individuals with higher cognitive function in childhood and adolescence tend to have longer years of education, progress to more professional and better-paid jobs, live healthier lives, and live longer, the direction of these genetic correlations of educational attainment or general cognitive function with schizophrenia was opposite (Table 2). The discrepancy may be derived from at least two disease subtypes: high intelligence and a cognitive disorder [70].

Comment 9:

The reviewer asked why Figure 2 was needed and whether the data of Okbay et al (2016) in Figure 2 can be included in Figure 3, Education, Educational Attainment. Figure 2 indicated genetic correlations between schizophrenia and educational attainment or general cognitive function, while Figure 3 indicated genetic correlations between general cognitive function and several phenotypes. Therefore, genetic correlation between schizophrenia and educational attainment, i.e., the data of Okbay et al (2016) in Figure 2, cannot be included in Fig 3.

Comment 10:

The reviewer suggested providing some statistical values to indicate whether correlations are meaningfully significant or just non-significant trends in Figure 3. According to the reviewer’s suggestion, we have indicated significant levels as follows:

Figure 3. Genetic correlations (rg) between general cognitive function and several phenotypes. Error bars indicate the SE of rg. *p<0.05, **p<0.01, ***p<0.001.< span="">

Moreover, the reviewer suggested describing when cognitive functions were measured in relation to the onset of psychiatric disorders. According to the reviewer’s suggestion, we have described it as follows:

P8, L79-81

As shown in Table 1, some individuals between early childhood and young adulthood were included in the correlation analyses. Considering the age range, the cognitive functions might be measured in some individuals before the onset of psychiatric disorders.

Comment 11:

The reviewer suggested avoiding abbreviation of "ID" for intellectual decline because the term "ID" is the standard abbreviation of intellectual disability. According to the reviewer’s suggestion, we have avoided using the abbreviation of "ID" for intellectual decline throughout manuscript.

Comment 12:

The reviewer pointed out that a GWAS analysis of n=166 was practically meaningless. The reviewer suggested not discussing such data. According to the reviewer’s suggestion, we have revised the sentence as follows:

P8, L100-102

To date, no large-scale GWAS for intelligence decline in patients with schizophrenia has performed [50]. Further studies to perform GWAS, PRS analysis or LDSC analysis using a larger sample size are needed.

Comment 13:

The reviewer suggested mentioning ‘IQ measures after the onset of schizophrenia include a non-specific consequence of schizophrenia or effects of medication and a mechanistically-linked cognitive decline preceding the onset of illness.’. According to the reviewer’s suggestion, we have mentioned it as follows:

P8, L118-P9, L121

However, we should pay attention to interpret IQ measures after the onset of schizophrenia because the IQ includes a non-specific consequence of schizophrenia or effects of medication and a mechanistically-linked cognitive decline preceding the onset of illness.

Comment 14:

The reviewer suggested describing ‘IQ scores might not be sensitive enough to detect deficits in specific cognitive domains that are more affected in schizophrenia than other domains.’, and pointed out that the observation that "preserved IQ" group should be interpreted with caution. According to the reviewer’s suggestions, we have mentioned it as follows:

P9, L121-123

In addition, IQ scores might not be sensitive enough to detect deficits in specific cognitive domains that are more affected in schizophrenia than other domains. Therefore, patients with preserved IQ may display deficits in specific cognitive domains.

Round  2

Reviewer 1 Report

The authors have addressed the reviewers comments. I do not have any more comments for the authors.

Author Response

We appreciate the prompt and helpful review.

Reviewer 2 Report

The authors are commended for addressing most of the points this reviewer raised previously. The revised manuscript is much improved. However, the authors are advised to address some outstanding issues.

1) The expression "the impairments are emphasized" is not clear. How about "worsen" instead of "are emphasized"?

2)  There are still inconsistent statements. At one point the authors sate "... impairments are emphasized after onset" and in another section, they state "cognitive impairments are stable over time after onset".

3) "poorly affected by current antipsychotic medications".  Do the authors mean "not really affected" or "exacerbated"?

4) "the reverse polygenic overlaps" is still not clear.

5) Page 6 L58-64. This reviewer still does not quite understand what the authors mean by this paragraph. The authors are advised to consult a native English speaker to polish this paragraph and the entire manuscript.

6)  When correlations are computed, some individuals were tested before the usual age of disease onset  and others were tested around and after disease onset. In other words, the sample contained cognitive functions whose later psychosis risk is not clear and cognitive impairments after disease onset.  That some studies included only some of these age groups. it is a rather problematic sample structure.

7) A GWAS study with 166 samples is not reliable and it is more misleading than meaningful. It (50) is still cited.

Author Response

We appreciate the prompt and helpful review and have revised the manuscript to accommodate the reviewer’s suggestions.

Response to Reviewer 2

Comment 1:

The reviewer suggested changing the expression ‘emphasized’ into ‘worsen’. According to the reviewer’s suggestion, we have revised it as follows:

P2, L54-55

These impairments exist before the onset of illness, and are worsened around it [20-22].

Comment 2:

The reviewer pointed out that there were inconsistent statements; at one point we stated ‘... impairments are emphasized after onset’ and in another section, we stated ‘cognitive impairments are stable over time after onset’. We have revised these statements as follows;

P2, L54-57

These impairments exist before the onset of illness, and are worsened around it [20-22]. It has been suggested that cognitive deficits of schizophrenia may be resistant to treatment with antipsychotic drugs [23-26], indicating a need for clarifications of the mechanisms underlying these conditions.

Comment 3:

The reviewer suggested clarifying the expression ‘poorly affected by current antipsychotic medications’. According to the reviewer’s suggestion, we have revised it as follows:

P2, L55-57

It has been suggested that cognitive deficits of schizophrenia may be resistant to treatment with antipsychotic drugs [23-26], indicating a need for clarifications of the mechanisms underlying these conditions.

Comment 4:

The reviewer pointed out that the sentence ‘the reverse polygenic overlaps’ was unclear. According to the reviewer’s suggestion, we have revised it as follows:

P6, L39-44

Polygenic overlaps between alleles of general cognitive function and schizophrenia risk have been examined [37,58]. On the basis of the polygenic risk scores (PRS) derived from GWASs, a set of alleles associated with lower general cognitive function predicted an increased risk of vulnerability to schizophrenia. Conversely, polygenic alleles associated with schizophrenia-related risks predicted lower cognitive functions, particularly general cognitive function, performance IQ, attention, and working memory [37,58-64].

Comment 5:

The reviewer suggested consulting a native English speaker to polish this paragraph ‘Despite the strong positive genetic correlation between general cognitive function and educational attainment (Figure 3) and the evidence that individuals with higher cognitive function in childhood and adolescence tend to have longer years of education, progress to more professional and better-paid jobs, live healthier lives, and live longer, the direction of these genetic correlations of educational attainment or general cognitive function with schizophrenia was opposite (Table 2). The discrepancy may be derived from at least two disease subtypes: high intelligence and a cognitive disorder [70].’ and the entire manuscript. According to the reviewer’s suggestion, we have consulted a native English speaker and revised the entire manuscript as well as these sentence as follows:

P6, L50-57

Studies using this method have consistently reported negative correlation between general cognitive function and schizophrenia-related risks, with rg of approximately -0.2 (Figure 2) [40-44]. Specifically, higher educational attainment is associated with lower schizophrenia risk [67], whereas lower educational attainment predicts worse premorbid function and worse outcomes [67]. These correlations would be reasonable in view of positive correlations between educational attainment and general cognitive function (Figure 3). However, recent studies found a positive correlation between educational attainment and schizophrenia (Figure 2) [56,68]. This discrepancy may be explained by at least two disease subtypes, i.e., patients with high intelligence and those with cognitive impairments [69].

Comment 6:

The reviewer pointed out that genetic correlation analyses included a problematic sample structure. As the reviewer pointed out, the correlation analyses included the issue. We have mentioned it as follows:

P8, L72-74

As individuals around adolescence were included in correlational analyses (Table 1), the timing of cognitive assessment, i.e. before or after onset of the illness, may have obscured the results from these analyses.

Comment 7:

The reviewer suggested removing a reference [50] because A GWAS study with 166 samples is not reliable and it is more misleading than meaningful. According to the reviewer’s suggestion, we have removed it.
